# Nazo, the *Drosophila* homolog of the NBIA-mutated protein–c19orf12, is required for triglyceride homeostasis

**Perinthottathil Sreejith**[1], **Sara Lolo**[1], **Kristen R. Patten**[1], **Maduka Gunasinghe**[1], **Neya More**[1], **Leo J. Pallanck**[2], **Rajnish Bharadwaj**[1]*

**1** Department of Pathology and Laboratory Medicine, University of Rochester Medical Center, Rochester, New York, United States of America, **2** Department of Genome Sciences, University of Washington, Seattle, Washington, United States of America

* Rajnish_bharadwaj@urmc.rochester.edu

**Data Availability Statement:** The raw data for RNAseq can be publicly accessed at Dryad. doi:10.5061/dryad.8sf7m0csb.

## Abstract

Lipid dyshomeostasis has been implicated in a variety of diseases ranging from obesity to neurodegenerative disorders such as Neurodegeneration with Brain Iron Accumulation (NBIA). Here, we uncover the physiological role of Nazo, the *Drosophila melanogaster* homolog of the NBIA-mutated protein–c19orf12, whose function has been elusive. Ablation of *Drosophila* c19orf12 homologs leads to dysregulation of multiple lipid metabolism genes. *nazo* mutants exhibit markedly reduced gut lipid droplet and whole-body triglyceride contents. Consequently, they are sensitive to starvation and oxidative stress. Nazo is required for maintaining normal levels of Perilipin-2, an inhibitor of the lipase–Brummer. Concurrent knockdown of Brummer or overexpression of Perilipin-2 rescues the *nazo* phenotype, suggesting that this defect, at least in part, may arise from diminished Perilipin-2 on lipid droplets leading to aberrant Brummer-mediated lipolysis. Our findings potentially provide novel insights into the role of c19orf12 as a possible link between lipid dyshomeostasis and neurodegeneration, particularly in the context of NBIA.

## Author summary

Lipids such as triacylglycerols comprise a class of biomolecules that serve important biological functions and are therefore fundamental to life. Thus, it is not surprising that defects in lipid biology cause multiple human diseases such as obesity and diabetes. As lipids perform crucial structural and functional roles in the animal brain, disruption of lipid metabolism is also implicated in various neurodegenerative diseases including Alzheimer's and Parkinson's disease. In this study, we decipher the function of the *Drosophila melanogaster* gene *nazo*, whose human counterpart–c19orf12, is mutated in a familial neurodegenerative disorder Neurodegeneration with Brain Iron Accumulation (NBIA). The function of this gene is largely unknown. *Drosophila* is a powerful genetic system that has informed us about various aspects of human biology. Using this organism, we show that *nazo* is required for maintenance of lipid droplets, cell organelles that store

**Funding:** RB was supported by a grant, funded jointly by NBIA Disorders Association (NBIADA, USA), Associazione Italiana Sindromi Neurodegenerative da Accumulo di Ferro (AISNAF, Italy), Hoffnungsbaum e.V. (HoBa, Germany), and Stichting Ijzersterk (The Netherlands). The funders had no role in study design, data collection and analysis, decision to publish, or preparation of the manuscript.

**Competing interests:** The authors have declared that no competing interests exist.

triacylglycerols and other lipids. Loss of this gene has significant impacts on the life span and health of the animal. Our study will provide the foundation for future investigations into the role of human c19orf12 and the cause of NBIA.

## Introduction

Triglycerides and their numerous lipid derivatives (e.g., fatty acids, phospholipids) are fundamental to life as they serve diverse biological functions such as providing a source of energy, acting as building blocks for cell membranes and modulating signaling pathways [1,2]. Therefore, proper intake, synthesis, storage, transport, and metabolism of triglycerides are highly regulated processes that have profound impacts on animal health. Defects in triglyceride hemostasis have been implicated in numerous diseases including obesity, diabetes, and neurodegenerative disorders [1,3–5].

Triglyceride metabolic pathways are highly conserved throughout evolution, and animal models such as *Drosophila melanogaster* have been instrumental in advancing our understanding of lipid homeostasis [2,6]. Like human intestine, *Drosophila* gut serves as a crucial hub for triglyceride homeostasis by orchestrating its absorption, storage, and distribution to other organs. Triglycerides derived from the food are converted into fatty acids in the gut by intestinal lipases. Fatty acids are absorbed by the enterocytes, which reconvert them into triglycerides and store them in intestinal lipid droplets [2]. In addition, de novo synthesis of fatty acids and triglycerides from glucose serves as an alternative source of triglycerides. For mobilization out of enterocytes, triglycerides are hydrolyzed into diacylglycerols, which are subsequently packaged into lipoprotein particles and transported to fat body and other tissues via hemolymph [2].

The biochemical pathway for triacylglycerol synthesis from fatty acids is well characterized, and involves the enzymes GPAT, AGPAT, PAP and DGAT, which are conserved in *Drosophila* [2]. Triacylglycerol (TAG) synthesis occurs between endoplasmic reticulum (ER) leaflets at discrete subdomains delineated by proteins like Seipin [7–10]. Subsequently, triglycerides are packaged into lipid droplets (LD), which bud off from the endoplasmic reticulum to form lipid droplets. A subset of lipid droplets maintain physical association with ER through endoplasmic reticulum-lipid droplet (ER-LD) contact sites [10]. In addition to influencing lipid droplet biogenesis, ER-LD contact site proteins can serve as a direct conduit for transfer of lipids and proteins between the organelles they bridge [11,12]. Their significance is underscored by the fact that mutations in genes encoding ER-LD contact site proteins like Seipin lead to a wide variety of deficits ranging from lipodystrophy to neurodegeneration [3,4].

Depending on cellular and organismal demands, triglycerides are converted into fatty acids either by cytosolic lipases (e.g., ATGL/brummer) or lipophagy [13,14]. Fatty acids in turn can be channeled into various pathways such as Beta oxidation for energy production or Kennedy pathway for synthesis of phospholipids [15]. In *Drosophila*, the lipase Brummer plays a key role in lipolysis, and not surprisingly, *brummer* mutants are characterized by aberrant triglyceride accumulation [13]. A critical regulator of Brummer activity is the lipid droplet protein Perilipin-2, which maintains normal triglyceride stores by antagonizing Brummer [16].

Both catabolic and anabolic steps in triglyceride metabolism are exquisitely regulated by a battery of hormones such as insulin like peptides (DILPs) and adipokinetic hormone [1,17,18]. In addition, gut microbiota influence triglyceride homeostasis through modulation of various signaling pathways including the innate immune response NF-κB pathway and insulin pathway [19–21]. Interestingly, recent evidence suggests that the *Drosophila* NF-κB

homolog Relish and its downstream effector–Sting can affect fat body triglyceride levels, independent of their roles in immune response [22,23].

The current study focuses on the *Drosophila* homologs of human *c19orf12* gene, which is mutated in a familial neurodegenerative disorder–Neurodegeneration with brain iron accumulation (NBIA) [24–26]. As the name of the disease suggests, NBIA patients show prominent neuronal loss accompanied by iron accumulation in basal ganglia. However, these patients also suffer from neuronal dysfunction/loss involving other areas of brain that do not exhibit significant iron accumulation, suggesting that the primary role of c19orf12 may be outside iron homeostasis. In line with this, other NBIA-mutated genes participate in diverse biological pathways including lipid metabolism, mitochondrial function, autophagolysosomal degradation and iron metabolism [24,26].

The biological role of c19orf12 is largely unknown. Human c19orf12 is a widely expressed gene, which is present in brain but strikingly enriched in adipose tissue (based on human protein atlas, S1D Fig). In cell culture, it localizes to endoplasmic reticulum and mitochondria [27]. In concordance with this, fibroblasts from patients with *c19orf12* mutations show increased sensitivity to oxidative stress and increased mitochondrial calcium in response to ATP stimulation [27]. In addition, a recent study has demonstrated that c19orf12 knockout in M17 neuronal cells causes cytosolic iron accumulation and increased sensitivity to ferroptosis [28]. The only studies on this gene at an organismal level has been performed in *Drosophila*. c19orf12 is evolutionarily conserved and *Drosophila* has two homologs of this gene named– *CG3740* and *nazo (CG11671)*. Using RNAi transgenic lines, Iuso et al. showed that depletion of fly c19orf12 homologs results in neurodegeneration and bang sensitivity [29], a commonly used phenotype in *Drosophila* to assess neuronal function [29]. In an unrelated context, Goto et al demonstrated that the fly c19orf12 homolog–Nazo is involved in NF-κB-mediated antiviral response [30].

In this paper, we identify a novel function for the *Drosophila* c19orf12 homolog–Nazo in lipid homeostasis. We demonstrate that *nazo* mutants exhibit extensive lipid droplet depletion in gut. Consequently, these mutants have a diminished lifespan and are sensitive to starvation. Nazo inhibits lipolysis by keeping the lipase activity of Brummer in check. These findings are particularly interesting considering the accumulating evidence linking lipid dyshomeostasis to NBIA and other forms of neurodegeneration. Our study potentially provides novel insights into the physiological function of c19orf12 and the molecular basis of NBIA.

## Results

### *Drosophila nazo* mutants show diminished longevity and functional deficits

As previously described, human c19orf12 gene has two *Drosophila melanogaster* orthologs– *CG3740* and *nazo* [29]. To investigate the role of *Drosophila* c19orf12 orthologs, we generated deletion mutants for these genes using CRISPR-CAS9 based genome engineering approach. The *CG3740* mutant allele consists of a 134-nucleotide deletion along with insertion of a small fragment of pHD-DsRed-attP plasmid. This most likely represents a null mutant as the deletion includes the transmembrane domain and in combination with the insertion introduces multiple frameshift mutations into the downstream open reading frame (Fig 1A). The *nazo* mutant has a 657-nucleotide deletion that includes the translation initiation site and majority of the open reading frame (Fig 1A). The absence of *nazo* transcript is confirmed by RT PCR (S1A Fig). Both *nazo* and *CG3740* homozygous mutant flies are viable. However, *nazo* as well as *nazo/CG3740* double mutants demonstrate a marked reduction in longevity with a median life span of about 39 days as compared to 59 days for the wild type flies (Fig 1B). The deletion

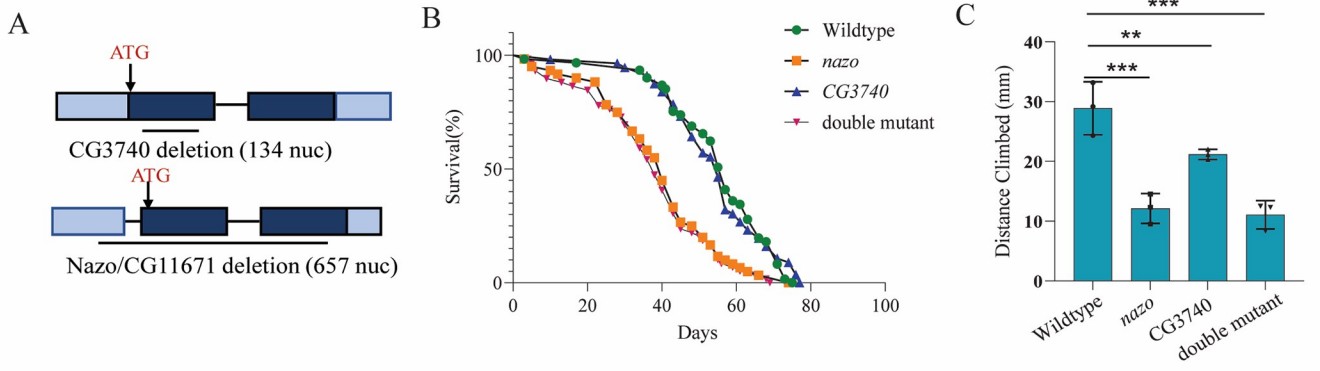

**Fig 1. Drosophila c19orf12 mutants show reduced lifespan, functional deficits, and dysregulation of lipid metabolism genes.** (A) Schematic of *Drosophila* Nazo and CG3740 genomic locus with deletion (open reading frame in dark blue, 5" and 3" untranslated regions in light blue, extent of deletion indicated by line) (B) Lifespan of males of indicated genotypes (N = 60; WT vs CG3740 p–ns, WT vs *nazo* p<0.0001, WT vs double mutant p<0.0001) (C) Standard climbing assay performed on 20-day old males of indicated genotypes. Data are mean ± s.e.m. ($N = 60$ for each genotype, ***$p < 0.001$ by Student's *t*-test). (D) Lipid metabolism genes upregulated or downregulated (with indicated fold change) in guts of 20-day old *nazo/CG3740* mutant males relative to WT flies.

of CG3740, on the other hand, has no significant effects on longevity (Fig 1B). To assess the functional significance of these mutations, we examined the locomotor abilities of these mutants using a standard climbing assay. At the age of 20 days, *nazo* single mutants as well as *CG3740/nazo* double mutants exhibit a markedly compromised locomotor ability (Fig 1C). The average distance climbed during the assay is 28.9 ± 1.7 mm for 20-day-old WT flies as opposed to 12.1 ± 1.0 mm and 11.1 ± 0.9 mm for *nazo* and *nazo/CG3740* double mutants, respectively (Fig 1C). The locomotor deficits in *nazo* mutants appear to be age-dependent as younger (5-days-old) *nazo* mutants did not show significant deficits in climbing assay (S1B Fig). Overall, these data demonstrate that the *Drosophila* c19orf12 ortholog Nazo is required for normal lifespan and fitness of the animal.

### *nazo* mutants exhibit gut lipid droplet depletion

According to modeENCODE tissue expression data, *CG3740* is variably expressed in multiple tissues (including brain and gut), whereas *nazo* is predominantly expressed in gut. We also confirmed by qRT-PCR based quantification that both *nazo* and *CG3740* are abundantly expressed in guts (S1C Fig). To gain insight into the physiological role of Nazo, we performed RNA seq analysis on wild type and *nazo/CG3740* double mutant male guts (where both these proteins are highly enriched). We identified 248 genes that were upregulated and 184 genes that were downregulated by more than two-fold in guts from nazo/*CG3740* double mutants relative to WT 20-day-old flies (S1 and S2 Files). Interestingly, many of the top hits from this analysis were genes involved in lipid metabolism (Fig 1D). This was particularly exciting as human c19orf12 is markedly enriched in adipose tissue (S1D Fig). Other cell biological

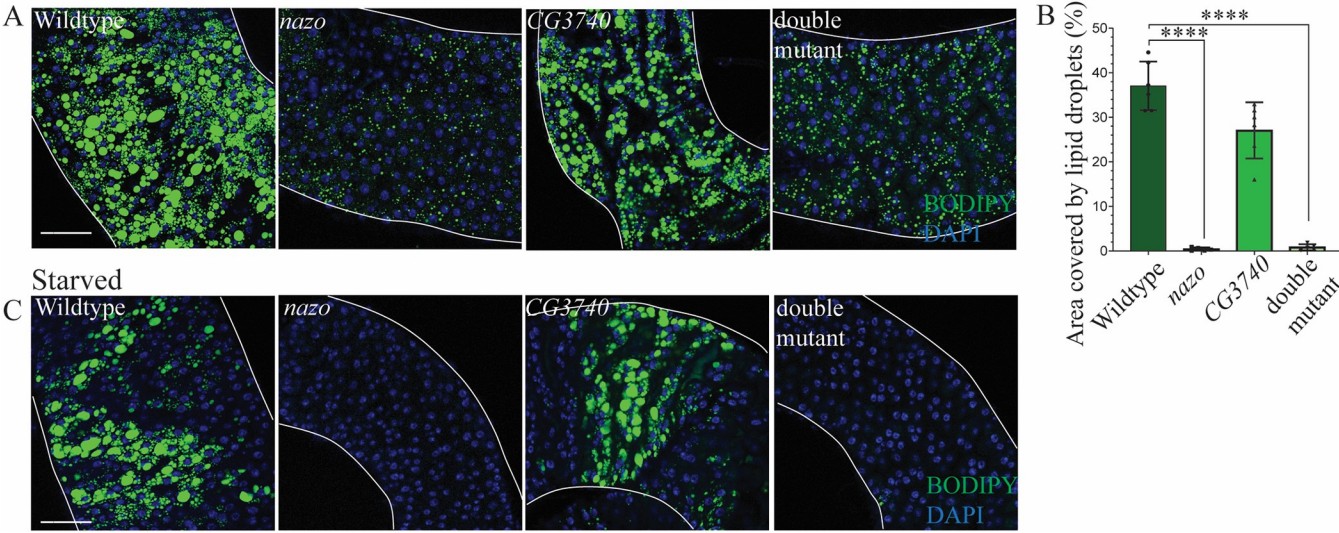

**Fig 2. *nazo* mutants show gut lipid depletion.** (A) Confocal images of guts from 20-day old male flies stained with BODIPY 493/503 dye show markedly reduced lipid droplet content in midguts. (BODIPY 493/503 dye–green and DAPI–blue; Scale bar = 10μm) (B) Quantification of percentage area occupied by lipid droplets in the midgut R4/5 region (N = 5–6 males; Student's t-test **** p-value = < 0.0001) (C) Guts from 20-day old male flies stained with BODIPY 493/503 dye after 40 hours of starvation show almost complete loss of lipid droplets in *nazo* and *nazo/CG3740* double mutants. (BODIPY 493/503 dye–green and DAPI–blue; Scale bar = 10μm).

processes significantly represented by the differentially expressed genes include metal homeostasis, proteases and transcriptional regulation (S3 File).

Prompted by this finding, we examined lipid droplets in *Drosophila* guts by BODIPY 493/503 dye staining. The distribution of lipid droplets is different between male and female guts. In male guts, lipid droplets are concentrated in two discrete areas, one in the anterior midgut and another in posterior midgut corresponding to the R4/5 region (S1E Fig). In contrast, in female guts, lipid droplets are dispersed diffusely throughout. We observed extensive lipid depletion in guts from *nazo* mutant and *nazo/CG3740* double mutant males relative to WT males (20-day-old) with 100% penetrance (Figs 2A, 2B, and S1E) in both LD-enriched areas. For the sake of consistency, all images presented, and quantifications are from the R4/5 region. Lipid droplet depletion is also present in mutant females but less pronounced than males (S1F and S1G Fig). Considering the more robust phenotype in males, subsequent studies were focused on males only. The severity and penetrance of this phenotype in *nazo* mutants is similar to *nazo/CG3740* double mutants, suggesting that among the two c19orf12 fly orthologs, Nazo plays the primary role in this context (Fig 2A and 2B). We also assessed the impact of starvation on this phenotype by examining the gut lipid content of 20-day-old WT and mutant males, after 40 hours of starvation. The gut lipid droplet depletion was even more striking in the *nazo* mutants and *nazo/CG3740* double mutants under starved condition (Fig 2C). Staining of wildtype and *nazo* guts at various ages, reveals an age-dependent decline in the LD content of mutant guts. Young flies (5-day-old) show relatively normal gut LD levels, whereas older flies reveal a dramatic loss of lipid droplets (S2A Fig). The loss of lipid droplet content is reflected not only in a reduction in LD numbers but also in their size (S2B Fig). Overall, our findings indicate that Nazo is required for maintenance of gut lipid droplets.

### *nazo* gut lipid depletion phenotype is not due to disrupted immune response

Multiple studies have shown that bacterial infection influences gut and fat body lipid droplet levels [21]. Furthermore, Nazo is transcriptionally induced by the fly NF-kB pathway and plays

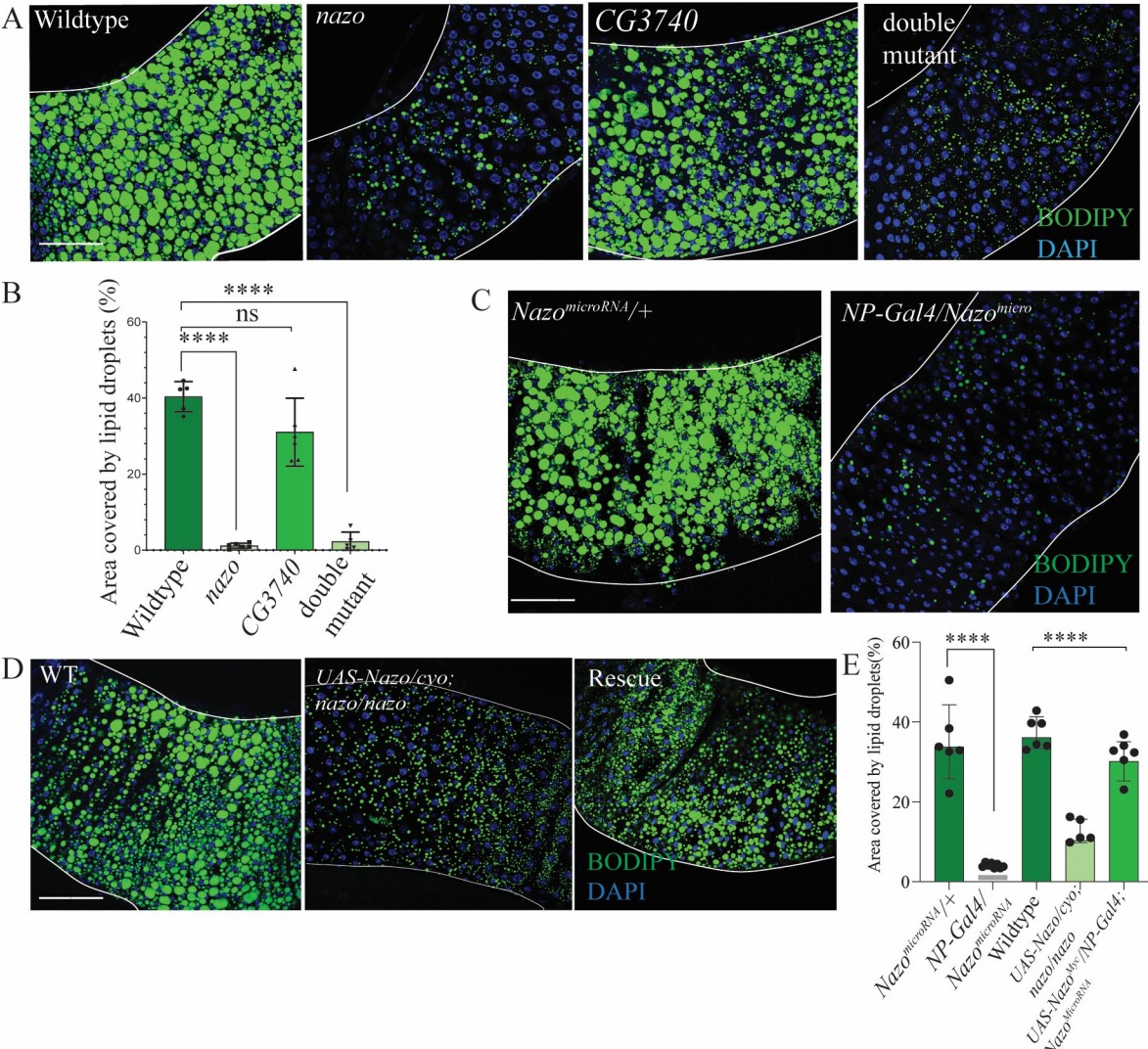

**Fig 3. *nazo* mutants grown in axenic conditions also show gut lipid droplet depletion. Nazo is required in enterocytes for gut lipid homeostasis.** (A) Guts from 20-day male old flies raised in axenic conditions show reduced levels of lipid droplets in *nazo* and *nazo/CG3740* double mutants compared to wildtype flies (BODIPY 493/503 dye–green and DAPI–blue; Scale bar = 10μm) (B) Quantification of percentage area occupied by lipid droplets in the midgut R4/5 region of axenic male flies of indicated genotype (N = 5–6 males; Student's t-test **** p-value = < 0.0001; ns = not significant) (C) Guts from 20-day old male flies expressing Nazo microRNA under enterocyte-specific *NP-Gal4* driver show reduced lipid droplets compared to control flies. (BODIPY 493/503 dye–green and DAPI–blue; Scale bar = 10μm) (D) Enterocyte-specific *NP-Gal4* driven expression of *UAS-nazo^{Myc}* transgene in *nazo* mutant background rescues the gut lipid droplet depletion phenotype. (BODIPY 493/503 dye–green and DAPI–blue; Scale bar = 10μm) (E) Quantification of the area covered by lipid droplets in the midgut R4/5 regions of indicated genotypes. (N = 5–6; Student's t-test **** p-value = < 0.0001).

a role in viral immune response in S2 cells [30]. Though gut lipid depletion was observed in *nazo* mutants in the absence of any exogenously introduced bacterial infection, we asked if this phenotype was related to impaired immune response. To this end, we performed lipid droplet staining on axenic (germ-free) WT and mutant flies generated through antibiotic treatment (31). We found that under axenic conditions, *nazo* and *nazo/CG3740* double mutant flies showed similar lipid droplet depletion phenotypes as seen in untreated condition (Fig 3A and 3B). Given the above-mentioned connection between NF-kB pathway and *nazo*, and the fat body lipid depletion phenotype observed in *relish* and *dSTING* mutants, we also performed

lipid droplet staining on guts from flies lacking these proteins [22,23]. Gut lipid droplet deple-
tion was not observed in *relish* mutants or in flies with *actin-Gal4* driven ubiquitous expression
of *dSTING* RNAi (S3A Fig). In fact, there was an increase in LD levels in these mutants relative
to WT flies. The *dSTING* RNAi phenotype was consistent with lack of LD loss phenotype in
the guts of *dSTING* mutant, as demonstrated by Akhmetova et al. [22]. Collectively, these find-
ings suggest that the lipid depletion phenotype observed in *nazo* mutants is not secondary to
compromised immune response.

### *Drosophila* Nazo is required in enterocytes for gut triglyceride homeostasis

The lipid depletion phenotype in *nazo* mutants can be due to a defect in feeding or absorption
of fat in the gut. To assess the feeding behavior, we performed a well-established feeding assay,
in which the flies are starved for two hours and subsequently provided food containing blue
dye [32]. The total amount of food consumed in 24 hours was calculated using colorimetry for
the blue dye. This assay did not reveal any defect in feeding behavior in the mutants (S3B Fig).
To test whether the absorption of fatty acids into the gut was affected in the mutant flies, we
fed adult flies with food containing fluorescently tagged oleic acid (BODIPY C16). 48 hours
after feeding, the guts were fixed, and examined by fluorescence microscopy. There was no
obvious difference in the levels of gut-incorporated fluorescent oleic acid between the wild-
type control and the mutants, suggesting that absorption of fatty acids was not defective in
*nazo* mutants (S3C Fig.). In addition, whole body glucose quantification did not reveal signifi-
cant differences in overall glucose levels in WT vs mutant flies (S3D Fig).

   To define the cell type in which nazo is required for the maintenance of gut lipid levels, we
examined the effect of selective depletion of *nazo* using various cell type specific Gal4 drivers.
To this end, we generated transgenic lines harboring two microRNAs in tandem, targeting
two distinct sequences in the *nazo* transcript [33]. Loss of *nazo* transcripts after microRNA
expression was confirmed by assessing the relative expression of *nazo* in gut RNA extracts
(S3E Fig). Depletion of nazo in enterocytes using the NP-Gal4 driver (Fig 3C and 3E) resulted
in dramatic loss of lipid droplets in the gut, at levels similar to the *nazo* null mutants, whereas
neuron and muscle-specific knock down using the Elav-Gal4, and Mef2-Gal4 drivers, respec-
tively had significantly milder effects on gut LD levels (S4A and S4B Fig). The subtle pheno-
types observed in flies with Elav-Gal4 or Mef2-Gal4 driven expression of microRNA is likely
due to leaky expression of these drivers. Using an alternative approach, we assessed if NP-Gal4
driven enterocyte-specific expression of Nazo transgene (*UAS-Nazo*) can rescue the *nazo*
mutant phenotype. A transgenic line that expresses *nazo* transcripts at levels similar to the
endogenous transcript (as detected by qRT-PCR) was chosen for these experiments (S5A Fig).
Indeed, expression of Nazo in enterocytes leads to restoration of lipid droplets in the guts of
*nazo* mutants (Fig 3D and 3E). Furthermore, enterocyte-specific rescue of Nazo also sup-
pressed the climbing defects, a measure of reduced fitness in these mutants (S5B Fig). Collec-
tively, these findings indicate that Nazo is required in enterocytes for maintenance of gut lipid
droplets, which is also consistent with the marked enrichment of the *nazo* transcript in the gut.

   Given the enterocyte-specific role for Nazo, we decided to examine if it influences the over-
all morphology, intestinal barrier integrity or cellular composition of guts. The size of guts is
normal in *nazo* mutants (S5C Fig). To assess the intestinal barrier function in *nazo* mutants,
we performed a well-established Smurf assay, which examines penetration of blue dye (admin-
istered through food) outside the digestive tract [34]. There was no significant difference in the
gut barrier integrity of *nazo* mutants relative to control flies (S5D Fig). To understand the
effect of *nazo* on the cellular architecture of the gut, we stained appropriately aged guts with
antibodies against Armadillo–an intracellular partner of E-cadherin, which labels enterocytes,

enteroblasts (EB) and intestinal stem cells (ISCs), and Sox21a, a transcription factor highly expressed in ISCs and EBs. Based on anti-Armadillo staining, we did not observe any obvious defects in the morphology or density of enterocytes (S5E Fig). Sox21a immunostaining reveals the presence of abundant enteroblasts and stem cells in both WT and *nazo* mutants. We also performed immunostaining for pH3 (phospho-Histone 3), a mitotic marker for the proliferative status of stem cells. This revealed a higher number of proliferating stem cells in *nazo* mutants (S5F Fig). This phenotype is age-dependent as younger 5-day-old *nazo* mutants show normal levels of ISC proliferation. Various intrinsic and extrinsic factors including stress, aging and metabolic alterations are known to influence ISC proliferation. Further studies are required to understand the role of Nazo in ISC proliferation and cellular stress.

### *Drosophila c19orf12* mutants have diminished triglyceride stores and are sensitive to starvation

Most energy reserves in flies are in the form of lipids droplets in the fat body. Given the gut lipid depletion phenotype of *nazo* mutants, we measured the lipid droplet content of fat bodies in 40-day old adult males by BODIPY 493/503 staining (Fig 4A and 4B). This revealed a significant reduction in fat body lipid droplet content in *nazo* mutants as well as *nazo/CG3740* double mutants compared to WT flies. Ubiquitous expression of *actin-Gal4* driven *nazo* microRNA also recapitulates this phenotype (S6A and S6B Fig). We also quantified the triglyceride levels from the whole bodies of the animals (Fig 4C). In line with the previous findings, we observed significant reduction in the total triglyceride content in the *nazo* mutant and *nazo/CG3740* double mutants compared to wildtype flies (Fig 4C). In this assay, *CG3740* mutants also show reduction in whole-body triglyceride content, suggesting that though the LD content of preserved fat bodies may be unaffected in *CG3740* mutants, overall mass of fat bodies may be reduced (Fig 4C). The reduction in organismal TAG levels could be rescued by enterocyte-specific expression of *nazo* using NP-Gal4 (Fig 4C). We also measured whole body TAG levels after NP-Gal4 driven enterocyte-specific or Lsp2-Gal4 driven fat-body specific expression of Nazo microRNA. TAG levels are significantly reduced when Nazo was depleted in enterocytes but not in fat bodies (S6C Fig). These experiments suggest that the function of Nazo in enterocyte but not in fat bodies is required for maintenance of organismal TAG levels.

Since *nazo* mutants show reduced lifespan along with diminished triglyceride levels, we asked whether these mutants are sensitive to starvation. To test this hypothesis, we subjected adult male flies to starvation stress by transferring them to vials containing wet filter paper without any other nutrients (Fig 4D). The percentage of dead flies were counted every 12 hours. This revealed that *nazo* mutants or *nazo/CG3740* double mutants are more sensitive to starvation (Fig 4D). *CG3740* mutants also showed increased starvation sensitivity in this assay, possibly due to reduced whole body TAG levels (Fig 4C). Though, *CG3740* mutants do not exhibit significant lipid droplet depletion in guts, the role of CG3740 in fat body remains to be examined further. In summary, these observations suggest that in addition to gut lipid depletion, *nazo* mutants exhibit diminished whole-body triglycerides stores, which manifests as vulnerability to starvation.

### *nazo* mutants are sensitive to oxidative stress and exhibit crop engorgement

The inability to sequester fatty acids in the lipid droplets results in oxidative stress [35]. Considering the diminished triglyceride stores in *nazo* mutants, we decided to examine if these mutants are more sensitive to oxidative stress. To this end, WT and mutant flies were transferred to vials containing wet filter paper with 5% $H_2O_2$ and the number of dead flies was

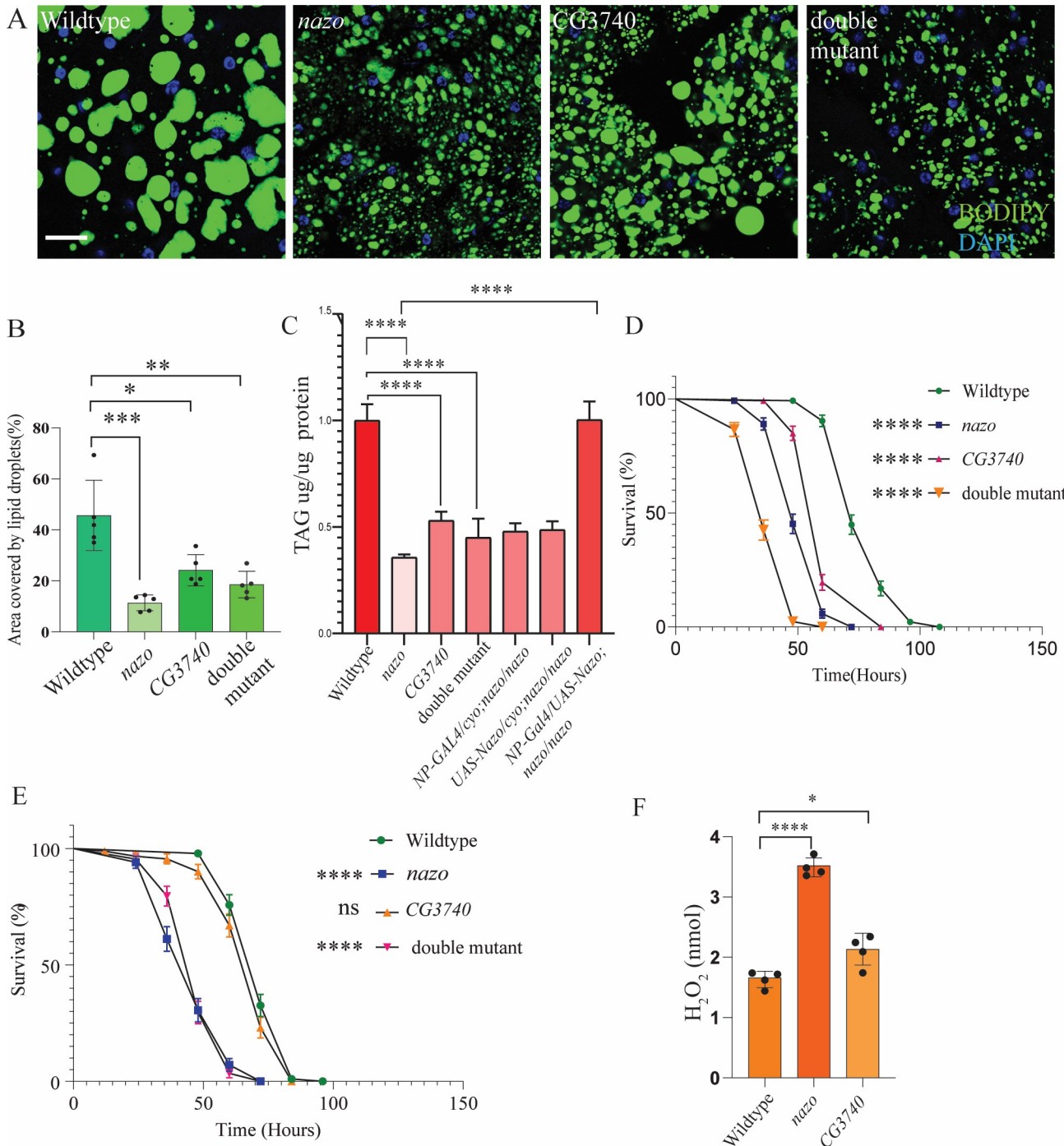

**Fig 4. *nazo* mutants show diminished organismal triglyceride stores along with increased sensitivity to starvation and oxidative stress** (A) Confocal images of fat-body of 40-day old male adults of indicated genotypes, stained with BODIPY 493/503 dye (BODIPY 493/503 dye–green and DAPI–blue; Scale bar = 10μm) (B) Quantification of percentage area covered by lipid droplets in each fat body (N = 5–6 males; Student's t-test, p-value ***$p < 0.0006$. * = 0.0129, ** = 0.0034) (C) Quantification of the TAG levels in whole body of 20-day old males with indicated genotypes. Rescue flies represent and *NP-Gal4::UAS-Nazo^{Myc}*; *nazo/nazo*. 6 males per genotype per biological replicates were used in triplicates. (Student's t-test **** p-value = $< 0.0001$) (D) Percentage of surviving male flies of indicated genotypes under starvation conditions, counted every 12 hours (N = 100), Student's t-test, p-value ***$p < 0.00001$ (E) Percentage of surviving male flies of indicated genotypes raised on food supplemented with 5% $H_2O_2$, counted every 12 hours (N = 80),Student's t-test, p-value ***$p < 0.00001$ (F) Whole body $H_2O_2$ quantification shows that *nazo* mutants along with *CG3740* have increased $H_2O_2$ levels relative to WT flies. 8 males per genotypes per biological replicates were used in quadruplets. (Student's t-test **** p-value = $< 0.0001$. * = 0.0188).

counted every 12 hours. Both *nazo* and *nazo/CG3740* double mutants revealed similarly increased sensitivity to $H_2O_2$ compared to wild type flies (Fig 4E). Prompted by this finding, we examined the total levels of $H_2O_2$ in whole bodies of 20-day-old WT and mutant males. We saw a considerable increase in overall $H_2O_2$ levels in *nazo* but not in *CG3740* mutants, compared to wildtype control (Fig 4F).

There was another intriguing finding in these mutants. When raised on high fat diet (30% coconut oil) but not on normal diet, *nazo* as well as *nazo/CG3740* double mutants develop a dramatic 5-fold enlargement of their crops on average (S6D Fig). Dye based gut motility assay revealed that this was due to diminished gut motility on high fat feeding conditions (S6E Fig). These findings raise the possibility that due to defects in triglyceride metabolism *nazo* mutants retain fat-rich food in their crops, though future studies are needed to understand the precise mechanism of food retention in *nazo* mutants. Together, these data demonstrate various impacts of lipid dyshomeostasis in *nazo* mutants including increased sensitivity to oxidative stress.

## The lipid droplet depletion in *nazo* mutants is due to dysregulated lipolysis

The lipid droplet depletion in *nazo* mutants can arise from decreased anabolism or increased catabolism of triglycerides. We first sought to address if fatty acids derived from triglycerides in the food can be properly absorbed and incorporated into the lipid droplets in *nazo* mutants. To test this possibility, we examined if high fat diet can rescue the *nazo* mutant phenotype. 20-day-old wild type and mutant flies were fed on a diet containing 30% coconut oil for 3 days and gut lipid droplet was examined by BODIPY 493/503 staining. While *nazo* mutants on normal diet demonstrated significant gut lipid droplet depletion, mutants placed on high-fat diet showed restoration of gut lipid droplet content (Fig 5A and 5B). We next asked if gut lipid droplets can be maintained in these flies if they are transferred back to normal food from high-fat diet. Interestingly, three days after transfer to normal diet *nazo* mutants redevelop lipid depletion phenotype (Fig 5A and 5B). Overall, these findings suggest that fatty acids from diet can be properly incorporated into lipid droplets in *nazo* mutants but likely due to increased catabolism these mutants are not able to maintain their triglyceride content.

To examine the significance of lipolysis for *nazo* mutant phenotype, we asked if depletion of lipases can suppress the gut lipid droplet depletion in *nazo* mutants. We generated a line–*NP-Gal4, Nazo* $^{microRNA}$/*Cyo* that lacks *nazo* in guts due to *NP-Gal4* driven expression of *nazo* microRNA. As mentioned above, this line shows the same level of gut lipid droplet depletion as the *nazo null* mutant. This line was crossed with *brummer* $^{RNAi}$ to examine the effect of concurrent depletion of this lipase on *nazo* mutant phenotype. Interestingly, loss of Brummer leads to rescue of the *nazo* mutant phenotype. We also examined gut lipid droplets in flies expressing *brummer* $^{RNAi}$ in wild-type background. These flies show only a subtle increase in gut lipid droplet levels compared to WT flies, whereas Brummer depletion leads to striking increase in gut LD levels of *nazo* mutants (Fig 6A and 6B).

One of the critical regulators of Brummer is Perilipin-2, a lipid droplet protein, which maintains lipid droplets by antagonizing Brummer [16,36]. Like Nazo, depletion of Perilipin-2 leads to diminished fat stores. Therefore, we decided to check if the level of Perilipin-2 is altered in *nazo* mutants. Western blot analysis of gut lysates revealed that Perilipin-2 levels are lower in *nazo* mutants relative to WT flies at the age of 20 days (Fig 6C and 6D). To rule out the possibility that this reduction in Perilipin-2 levels is simply a reflection of diminished lipid droplets in *nazo* mutants, we examined Perilipin-2 levels in 7-day-old flies, which have abundant gut lipid droplets (S7A and S7B Fig). Young *nazo* mutants also exhibit reduced gut Perilipin-2, indicating that Perilipin-2 loss in these mutants precedes and thus may play a causal

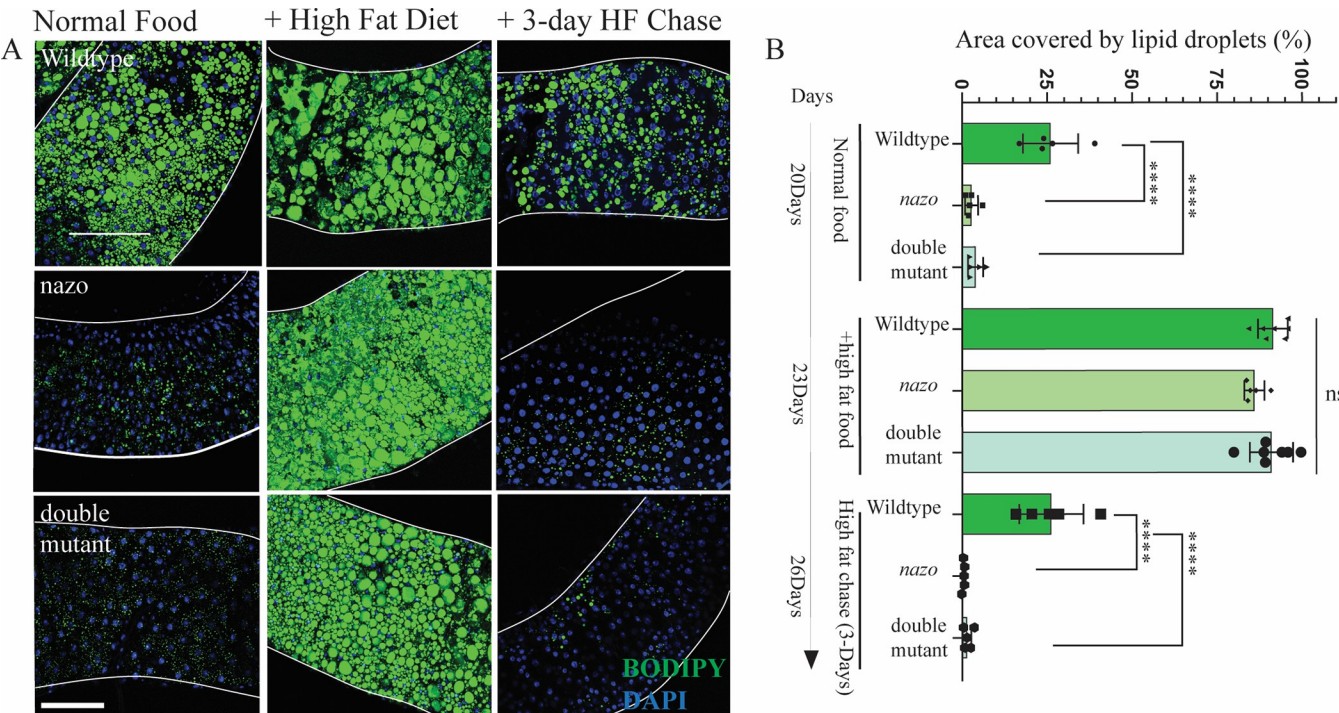

**Fig 5. High-fat diet can rescue gut lipid loss in *nazo* mutants.** (A) Confocal images of BODIPY 493/503 stained guts from male flies of indicated genotypes raised on normal diet or high-fat diet for 3 days or transferred to normal diet for 3 days after high-fat diet treatment; high-fat diet rescues gut lipid droplet depletion in *nazo* and *nazo/CG3740* double mutants, but the phenotype reappears when flies are transferred back to normal diet (Scale Bar 10μm). (B) Quantification of the percentage area occupied by lipid droplets in indicated genotypes and conditions, (N = 5–6 males, Student's t-test **** p-value = < 0.0001).

role in lipid droplet depletion (Fig 6C and 6D). These findings were also recapitulated by immunostaining experiments on guts of flies harboring a *GFP-trap* in the endogenous *perilipin-2* locus (*Lsd-2^CPTI002546^*). GFP immunostaining reveals that Perilipin-2 localizes around lipid droplets in guts. This staining is dramatically reduced when Nazo is concurrently depleted by microRNA (S8A Fig).

These findings prompted us to test if *perilipin-2* overexpression can rescue the lipid depletion phenotype of *nazo* mutants. As discussed earlier, microRNA mediated depletion of *nazo* results in loss of lipid droplet in the gut (Fig 6A). Co-expression of *perilipin-2* in this setting, however, leads to suppression of this phenotype (Fig 6A). We also repeated the above-mentioned experiments using the ubiquitous Gal4-driver *actin-Gal4 in CG3740 mutant; nazo microRNA* background. The results are essentially similar in both set of experiments, consistent with the enterocyte specific role of Nazo. In this experimental set up, we also tested the impact of depletion of two other lipases Dob and CG5966 as well as overexpression of Sturkopf, another lipid droplet associated protein. Unlike *brummer* RNAi or *perilipin-2* overexpression, manipulation of *dob*, *CG5966* or *sturkopf* did not suppress the *nazo* mutant phenotype (S9A and S9B Fig). Overall, these experiments provide a plausible explanation for lipid droplet defect in *nazo* mutants. These mutants exhibit diminished Perilipin 2, which in turn may result in increased lipolysis due to overactivity of the lipase Brummer.

In summary, we show that the *Drosophila* c19orf12 homolog–Nazo maintains gut lipid droplets by regulating lipolysis. Given the role of lipid dyshomeostasis in neurodegeneration, our findings provide the foundation for future studies on the role of mammalian c19orf12 in lipid homeostasis and neurodegeneration.

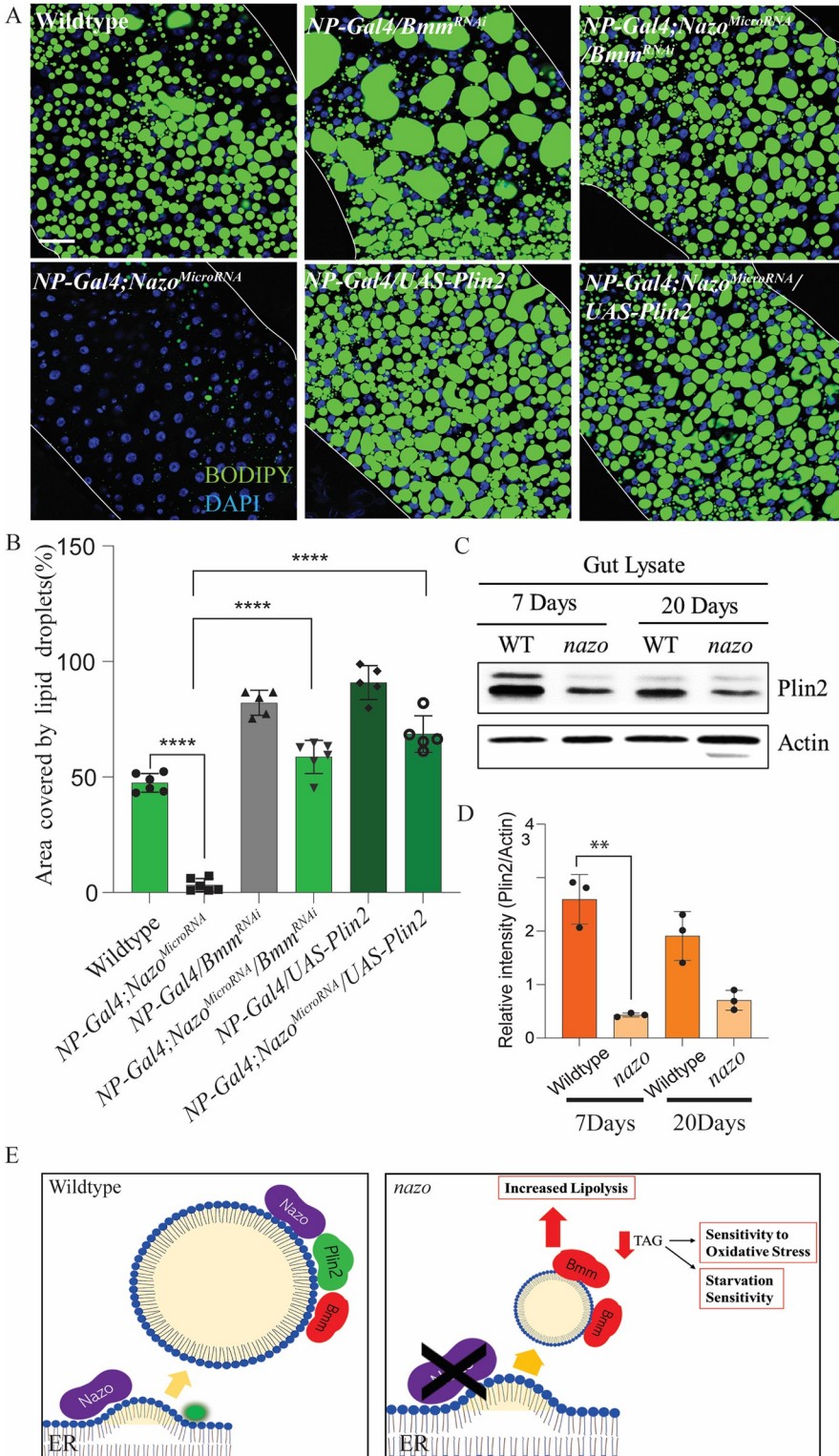

**Fig 6. The lipid droplet phenotype in *nazo* mutant is due to dysregulated lipolysis.** (A) Confocal images of male guts from 20-day old male flies of indicated genotypes stained with BODIPY 493/503 dye (BODIPY 493/503 dye–green and DAPI–blue; Scale bar = 10μm); Concurrent enterocyte specific knockdown of the lipase Brummer rescues the reduced gut lipid droplet phenotype resulting from Nazo depletion. Similarly, overexpression of Perilipin-2, a regulator of Brummer, rescues the Nazo depletion phenotype. (B) Quantification of the percentage area occupied by

lipid droplets midguts of flies of indicated genotypes (N = 5–6 males, Student's t-test **** p-value = < 0.0001). (C) Immunoblot showing reduced expression of Perilipin -2 in gut lysate from 20-day and 7-day old *nazo* mutants relative to wild type flies, (D) Quantification of the relative intensity of Plin2 /Actin from western blots. (Student's t-test **** p-value = < 0.0013) (E) Model for the role of Nazo in lipid homeostasis. Nazo is a lipid droplet associated protein that maintains lipid droplets by restraining the lipase activity of Brummer. Lack of Nazo leads to increased lipolysis and reduced TAG levels resulting in increased sensitivity to oxidative stress and starvation.

## Discussion

Multiple lines of evidence point to a role for c19orf12 in lipid homeostasis. First, human adipose tissue and other adipocyte-rich organs such as breast and liver show striking enrichment of c19orf12 (S1D Fig, spliced from RNA expression consensus dataset from human protein atlas). Second, ablation of the fly c19orf12 homologs results in differential expression of multiple lipid metabolism genes in guts. Finally, and most importantly, flies lacking the c19orf12 homolog–Nazo exhibit a robust and completely penetrant lipid droplet depletion phenotype in their guts. Nazo plays a cell autonomous role in enterocytes to maintain the gut lipid droplet pool. This phenotype is not exacerbated by the added absence of CG3740, suggesting that Nazo is the primary player in this context. It is intriguing that even though CG3740 is expressed more widely than Nazo, its impact on the overall fitness of the organism (as assessed by locomotor and viability assays) is significantly less pronounced than Nazo. The function of CG3740 is still unclear and future studies are needed to investigate its role in neuronal and nonneuronal tissues. In summary, the expression pattern of c19orf12 in humans and the physiological role of Nazo in *Drosophila* is suggestive of a conserved role for this protein in lipid homeostasis.

What are the functional consequences of gut triglyceride depletion observed in *nazo* mutants? Disruption of *nazo* has profound impact on the health of animals as evidenced by reduced life span and locomotor deficits in these mutants. Triglyceride stores are particularly important for the sustenance of animals during phases of starvation. Thus, as expected, *nazo* mutants are highly vulnerable to starvation. In addition to fulfilling the energetic needs of the animal, lipid droplets also play a protective role against reactive oxygen species by sequestering fatty acids into triglycerides and preventing aberrant lipid peroxidation. In concordance with their lipid droplet depletion phenotype, ablation of *nazo* leads to increased sensitivity to hydrogen peroxide induced oxidative stress. Thus, compromised lipid metabolism in *nazo* mutants has significant implications for the sustenance of animals. RNAseq analysis reveals altered expression of many lipid metabolism genes in *nazo* mutants. These include genes involved in triglyceride transport/metabolism such as CG31741, the fly homolog of the HDL-receptor SCARB1, CG30054 a putative lipid transporter and various acyl-CoA synthetases. These proteins may allow *nazo* mutants to cope with gut and organismal triglyceride depletion. In addition, many of the genes involved in cholesterol and steroid metabolism are also differentially expressed in these mutants suggesting that Nazo may be involved in homeostasis of other lipid classes also.

*nazo* was identified as one of the genes upregulated in response to viral infection in a NF-κB dependent manner [30]. Innate immune response is intricately linked to metabolism and multiple studies have shown that bacterial infection and innate immune response effect gut and fat body triglyceride levels [19–21]. While it is plausible that Nazo serves as a link between immune response and metabolism by modulating triglyceride homeostasis, our findings suggest that Nazo governs lipid homeostasis under physiological conditions, independent of its possible role in immune response. We observe lipid droplet depletion in *nazo* mutants in normal rearing conditions in the absence of extraneous bacterial infection. More importantly, this phenotype is observed in axenic flies, which are raised in the presence of antibiotics and lack

gut microbiota indicating that gut commensal bacteria are not responsible for this phenotype. Interestingly, two studies have recently shown that NF-κB and dSTING influence triglyceride metabolism in an immune response-independent fashion. While *NF-κB (relish)* mutants exhibit a rapid decline in fat body triglyceride stores on starvation, *dSTING* mutants show diminished triglyceride stores in fat body even on normal diet [22,23]. These studies prompted us to examine the gut lipid droplet levels in *relish* mutants or *dSTING* RNAi flies. Unlike *nazo* mutants, *relish* mutants or *dSTING* RNAi flies show enlarged rather than diminished gut lipid droplets under normal rearing conditions. The latter observation is consistent with the report from Akhmetova et al. who did not observe reduced gut lipid droplet content in *dSTING* mutants [22]. Overall, our findings suggest that the lipid droplet depletion observed in *nazo* mutants is likely not due to disruption of immune response or dysregulation of NF-κB pathway.

Many of our observations support that increased lipolysis is responsible for lipid droplet depletion in *nazo* mutants. First, *nazo* mutant phenotype can be suppressed by feeding high fat diet to flies. When these flies are transferred back to normal diet, their lipid stores are depleted within a few days. Second, simultaneous depletion of the lipase–Brummer, rescues the gut lipid depletion induced by suppression of Nazo. Third, Perilipin-2, an inhibitor of Brummer is diminished in *nazo* mutants suggesting that Brummer overactivity, at least in part due to Perilipin-2 loss, may be responsible for triglyceride depletion in *nazo* mutants. Finally, overexpression of Perilipin-2 rescues the *nazo* mutant phenotype. Overall, these findings suggest that Nazo regulates lipolysis and maintains gut triglyceride levels at least in part through its effect on Brummer-Perilipin 2 axis. At least in S2 cells, we do not detect a robust physical interaction between tagged-Nazo and endogenous Perilipin-2 (S4 File). Further investigations are needed to assess potential physical interactions between these proteins or to identify novel direct interactors of Nazo that may serve as the link between these proteins. Mammalian c19orf12 localizes to endoplasmic reticulum and mitochondria [27]. Future studies, examining the colocalization of c19orf12 and Nazo with lipid droplets as well contact sites between LD and other organelles like endoplasmic reticulum and mitochondria will provide additional mechanistic information about the role of Nazo. In the absence of ER-LD contact site proteins like Seipin, lipid droplets show significant morphological abnormalities and alterations in their proteome [11,12]. In mammals, Seipin deficiency leads to increased lipolysis and marked loss of adipocytes, which can be partially rescued by the suppression of the lipase ATGL that is homologous to *Drosophila* Brummer [37,38]. Thus, disruption of LD biogenesis can eventually lead to LD loss due to increased triglyceride catabolism. Considering these observations, it is also possible that unrestrained lipolysis in *nazo* mutants is a secondary consequence of aberrant LD biogenesis.

Our findings suggest that Nazo's role at the lipid droplets is responsible for its mutant phenotypes. However, there are factors like sex and age that influence the severity of this phenotype. Gut lipid droplet depletion is present in *nazo* mutant females but less pronounced than the males. Sexual dimorphism in metabolic pathways is a topic of intense investigation and studies in *Drosophila* have been particularly illuminating [39–41]. It is well established that *Drosophila* males have less body fat than females. This can be attributed to increased levels of the lipase Brummer in adipose tissue and increased activity of the adipokinetic hormone secreting APC cells in the brain [40,41]. Thus, it is conceivable that due to above-mentioned factors, males are more susceptible to perturbations like Nazo loss, which enhance lipolysis. Nazo phenotypes also worsen progressively with age. Karpac et al. have demonstrated that aging wild type flies have diminished gut lipid droplets [17]. This is due to chronic activation of Foxo, which in turn represses expression of the intestinal luminal lipase magro. Diminished magro leads to compromised lipolysis of triglycerides in gut lumen and reduced uptake of

fatty acids by enterocytes [17]. Considering these findings, the disruption of the metabolic adaptation in aging flies may contribute to age dependent worsening of *nazo* phenotypes.

Iron accumulation in basal ganglia is a prominent feature of NBIA caused by c19orf12 loss [25]. Previous study from Iuso et al. did not reveal any iron accumulation in the brains of flies lacking c19orf12 homologs, a finding that we have confirmed [29]. Given the known role of iron on lipid metabolism, we cannot entirely exclude the contribution of iron metabolism to Nazo related lipid phenotypes. However, we favor a direct role for this protein at lipid droplets, which is necessary for the maintenance of this organelle. Though extensive iron accumulation is present in the basal ganglia of NBIA patients, it is unclear whether iron accumulation is the primary cause of neurodegeneration, or if it is an epiphenomenon. In NBIA patients, neuronal dysfunction/loss and functional abnormalities are observed in parts of brain other than basal ganglia where iron accumulation is not prominent. Furthermore, only two of the NBIA-mutated genes are directly involved in iron metabolism, whereas the others function in diverse processes including lipid homeostasis, mitochondrial function and autophagolysosomal degradation [26]. Though, it has been challenging to come up with a unifying mechanism for this disease, there is substantial evidence to suggest that disruption of lipid metabolism plays a key role in the pathogenesis of this disease. Two of the NBIA-mutated genes PLA2G6 and FAAH are directly involved in lipid metabolism. Two other genes–COASY and Pantothenate kinase are crucial components of the Coenzyme A biosynthesis pathway, a critical regulator of various aspects of lipid metabolism [26]. In continuation with this theme, our study provides the first evidence for the involvement of the fly c19orf12 homolog–Nazo in lipid homeostasis. Our findings provide a foundation for future studies, seeking to decipher the role of mammalian c19orf12 in neuronal and nonneuronal tissues, especially in the context of lipid biology.

The robust gut phenotypes of *Drosophila nazo* mutants and enrichment of c19orf12 in adipose tissue raise some interesting questions about cell non-autonomous effects of c19orf12 loss. Can lipid dyshomeostasis in nonneuronal tissues contribute to neurodegeneration in human NBIA patients? Lipid droplets are sites of detoxification/sequestration for various lipophilic xenobiotics and endogenous toxins. In addition, they are known to be protective against oxidative stress by sequestering fatty acids and preventing their peroxidation. Thus, defects in lipid droplet rich tissues including liver or adipose tissue can have deleterious systemic consequences and potentially affect the brain. Future studies are needed to explore these questions.

In summary, this study establishes a role for the c19orf12 homolog–Nazo in lipid homeostasis and provides novel insights into the mechanism of NBIA. Our findings will pave the way for exploring the connection between lipid dyshomeostasis, neurodegeneration and synuclein accumulation.

## Materials and methods

### Fly husbandry and genetics

The flies were reared in standard cornmeal/agar medium supplemented with yeast at 25˚C with 60 ± 5% relative humidity and 12h light /dark cycles unless specified. The following strains were obtained from Bloomington Drosophila Stock Center: *w^1118^*, *Actin-Gal4* (25374), *NP-Gal4*, *elav-Gal4*, *Lsp2-Gal4(84327)*, *Col4a1-Gal4(7011)*, *rel^E20^*, *brummer^RNAi^* (25926), *dob^RNAi^* (65925), *dSTING^RNAi^*, *CG5966^RNAi^* and *UAS-sturkopf* (91394). *UAS-Perilipin2* [13] was a kind gift from Dr. Ronald Kuhnlein. *Lsd-2^CPTI002546^* was a kind gift from Dr. Michael Welte.

The following lines were generated for this study *CG3740 null*, *nazo (CG11671) null*, *UAS-CG11671^myc^* and *pUAST-CG11671^microRNA^*. All the mutants used in this study were isogenized with the *w1118* strain before using them for any experiments.

*CG3740* and *CG11671 (nazo)* knockout allele was created by CRISPR/Cas9-mediated gene editing according to a published procedure [42]. Briefly, we replaced part of the coding sequence of these genes with DsRed through homology-mediated repair. Two different guide RNAs were used for each of these genes. Sequences flanking the guide RNA target sites were amplified from genomic DNA and introduced into donor plasmids to facilitate homology-directed repair.

The following primer sequences were used for CG3740 guide RNAs:
5'-Guide RNA

Sense oligo: 5'- CTTCGGAGGCGATAGCCATCGTAG -3'

Antisense oligo: 5'- AAACCTACGATGGCTATCGCCTCC -3'

3'-Guide RNA

Sense oligo: 5'- CTTCGCGGTTGGCGGGGCCGCCGG -3'

Antisense oligo: 5'- AAACCCGGCGGCCCCGCCAACCGC -3'

The following primers were used to generate homology arms for CG3740:
5'-Homology arm

Forward: 5'- ATATCACCTGCATATTCGCCAGGTACACCCATACACTAG -3'

Reverse: 5'- ATATCACCTGCATATCTACCGATGGCTATCGCCTCCATC -3'

3'-Homology arm

Forward: 5'- ATATGCTCTTCATATCGGCGGAATCGCCGCCTACAAGA -3'

Reverse: 5'- ATATGCTCTTCAGACTAGGTTATTCCATTACCATTCGG -3'

The following primer sequences were used for CG11671 guide RNAs:
5'-Guide RNA

Sense oligo: 5'- CTTCGTAACTCCCCTCCACAGTGG -3'

Antisense oligo: 5'- AAACCCACTGTGGAGGGGAGTTAC -3'

3'-Guide RNA

Sense oligo: 5'- CTTCGGGAATGACCATCGTTGATT -3'

Antisense oligo: 5'- AAACAATCAACGATGGTCATTCCC-3'

The following primers were used to generate homology arms for CG11671:
5'-Homology arm

Forward: 5'- ATATCACCTGCATATTCGCACTCGTGCCAACACTTCTTG -3'

Reverse: 5'- ATATCACCTGCATATCTACCTGTGGAGGGGAGTTACTCG -3'

3'-Homology arm

Forward: 5'- ATATGCTCTTCATATATT AGGCATTAATAGAAATA -3'

Reverse: 5'- ATATGCTCTTCAGACTCACTATGTTCTTATAAAAC -3'

Both 5' and 3' homology arms were then cloned into the pHD-DsRed-attP vector containing the eye-specific 3xP3 promoter fused with DsRed, and the resulting construct was microinjected into Cas9-expressing embryos by a commercial service (Rainbow Transgenic Flies Inc.).

Flies bearing the *CG3740 or CG11671* deletion were identified by screening the offspring of injected adults for expression of red fluorescence in the eye. The deletion mutants were confirmed by PCR and sequencing, using primers against a dsRed sequence and adjacent CG3740/CG11671 genomics sequences. Subsequently, DsRed markers inserted at the deletion sites were removed by crossing the mutants with hsCre flies, and maintaining the progenies at 25˚C. In *CG3740* mutants a small fragment of pHD-DsRed-attP plasmid persists at the deleted locus.

*CG11671* microRNA constructs targeting the sequences TTCAGCCATCTCAGAGATAAT C and GGTCATTTTGAACGATT TGACG in the nazo transcripts were generated as described [33]. Briefly, while keeping the stem-loop backbone and surrounding sequences unaltered, the *Drosophila* Mir6.1 gene-targeting microRNA sequence was replaced with the 22-bps complementary to the *nazo* transcript. Two microRNA constructs targeting distinct *nazo* sequences were inserted in tandem into the pUAST vector and the resulting plasmid was used to generate *UAS-Nazo^{microRNA}* transgenic flies. For rescue experiments, the *Drosophila nazo* transgene was cloned into pUAST vector along with a carboxy-terminal 3 X MYC tag.

Unless specified, all experiments were performed on 20-day-old males from different genotypes.

## Life span assay

For life span analysis 60 males for each genotype were transferred to fresh food (20 flies per vial) every 2–3 days and dead flies were counted. Statistical analysis was performed using log rank test.

## Climbing assay

Climbing behavior was examined by Rapid Iterative Negative Geotaxis (RING) assay at 5 and 20 days age, according to a previously published protocol [43]. Briefly, flies were transferred into plastic vials and six vials at a time were loaded onto the RING apparatus (10 flies per vial). The vials were tapped down to initiate the climbing response and the height climbed by each fly after 4 seconds was recorded. The climbing assay was repeated three times and the average height climbed in the three trials was calculated.

## RNA seq analysis

Total RNA was isolated from triplicates of 20 freshly dissected guts of 20-day old WT and *CG3740/nazo* mutant males using the RNeasy Plus Mini Kit (Qiagen, Valencia, CA) per manufacturer's recommendations. RNA-seq analysis was performed at UR Genomics Research Center. The total RNA concentration was determined with the NanopDrop 1000 spectrophotometer (NanoDrop, Wilmington, DE) and RNA quality assessed with the Agilent Bioanalyzer (Agilent, Santa Clara, CA). The TruSeq Stranded mRNA Sample Preparation Kit (Illumina, San Diego, CA) was used for next generation sequencing library construction per manufacturer's protocols. Briefly, mRNA was purified from 200ng total RNA with oligo-dT magnetic beads and fragmented. First-strand cDNA synthesis was performed with random hexamer priming followed by second-strand cDNA synthesis using dUTP incorporation for strand marking. End repair and 3'adenylation was then performed on the double stranded cDNA. Illumina adaptors were ligated to both ends of the cDNA, purified by gel electrophoresis and amplified with PCR primers specific to the adaptor sequences to generate amplified libraries of approximately 200-500bp in size. The amplified libraries were hybridized to the Illumina flow cell and single end reads of 75nt were generated for each sample using Illumina's NextSeq550 sequencer.

Analysis methods: Raw reads generated from the Illumina basecalls were demultiplexed using bcl2fastq version 2.19.0. Quality filtering and adapter removal are performed using FastP version 0.20.0 with the following paramters: ""—length_required 35—cut_front_window_size 1—cut_front_mean_quality 13—cut_front—cut_tail_window_size 1—cut_tail_mean_quality 13—cut_tail -y -r". Processed/cleaned reads were then mapped to the D. melanogaster reference genome (BDGP6.28) using STAR_2.7.0f with the following parameters: "—twopass Mode Basic—runMode alignReads—outSAMtype BAM SortedByCoordinate–outSAMstrandField intronMotif—outFilterIntronMotifs RemoveNoncanonical–outReads UnmappedFastx". Genelevel read quantification was derived using the subread-1.6.4 package (featureCounts) with a GTF annotation file and the following parameters: "-s 2 -t exon -g gene_name". Differential expression analysis was performed using DESeq2-1.22.1 with a P-value threshold of 0.05 within R version 3.5.1 (https://www.R-project.org/). A PCA plot was created within R using the pcaExplorer to measure sample expression variance. Heatmaps were generated using the pheatmap package were given the rLog transformed expression values. Gene ontology analyses were performed using the EnrichR package.

The raw data for RNAseq is accessible from Dryad as follows [48]. RNA sequencing data from the guts of Drosophila wildtype and CG3740/nazo double mutant 20-day-old males [Dataset]. Dryad. https://doi.org/10.5061/dryad.8sf7m0csb

## qRT-PCR

RNA was isolated from ten 20-Day old male midguts using Direct-zol RNA miniprep with TRI reagent (R2081, Zymo research, with DNase1 treatment) according to the manufacturer's instruction. cDNA was synthesized with 1ug of total RNA using Superscript-III reverse transcriptase (Invitrogen). Real-time PCR was performed using the Applied biosystems QuantaStudio 5 using Quantabio Perfecta SYBR green mix according to manufacturer's protocol using three independent biological replicates, using the following primers:

*Nazo* Forward: 5' AACGATTTGACGGAGAGTCAG 3',
*Nazo* Reverse: 5' CTTGCACATGCCGGTTATTTAG 3',
*CG3740* Forward: 5' TGGGCGAAGTCATCCTAAAC 3',
*CG3740* Reverse: 5' CACGTCGGTGGGATGTATG 3',
*rp49* Forward: 5' CCAGTCGGATCGATATGCTAAG 3',
*rp49* Reverse: 5' CCGATGTTGGGCATCAGATA 3'.

Relative expression was calculated using the ΔΔCT method and normalized to *rp49* levels. P values were calculated using unpaired two-tailed student's t-test.

## Starvation and Oxidative stress sensitivity assay

For starvation analysis, 100 flies were transferred from normal food to vials containing Whatman filter paper soaked with PBS (20 flies per vial). Fresh PBS was added every 24 hours to prevent drying. Dead flies were counted every 12 hours.

For Oxidative stress 80 flies were transferred from normal food to vials containing 5% $H_2O_2$ (20 flies per vial). Dead flies were counted every 12 hours.

## Generation of axenic flies

Flies were reared in axenic conditions as described previously [31]. Briefly, embryos were washed in 2.5% chlorine (50% bleach) for 2 minutes followed by 2-minute wash in 70% alcohol and 2-minute wash in distilled water. The sterile embryos were transferred to food containing antibiotics (Tetracycline, chloramphenicol, ampicillin, and erythromycin). Axenic flies were transferred to fresh food supplemented with antibiotics every 2 days, in triplicates. Axenic

conditions were tested by grinding the flies in PBS and spreading the suspension on LB, MRS or Mannitol plates at 37˚C or 30˚C. Absence of bacterial colonies indicated that the flies were devoid of gut microbes.

### BODIPY 493/503 staining

Appropriately aged guts were dissected in PBS followed by fixation in 4% paraformaldehyde in PBS for 30 mins. The guts were incubated in BODIPY 493/503 (1ug/ml, Thermo fisher Scientific) for 30 minutes in dark with DAPI. The guts were washed with PBS for 30 mins and then mounted on slides in 70% glycerol. The guts were imaged immediately by confocal microscopy. For BODIPY 493/503 staining of fat bodies, adult carcasses were filleted in PBS, fixed in 4% paraformaldehyde and incubated in BODIPY 493/503 for 30 minutes.

### Triglyceride quantification

TAG quantification was performed using Triglyceride Quantification kit (MAK266, Sigma-Aldrich). For TAG quantification of whole body, 8 males were used in triplicates for each sample. Triglyceride standard solution and glycerol standard solution were used as standards. Briefly, whole males were homogenized in 100μl solution of 5% NP40 solution and heated for 2 min at 80–100˚C twice followed by centrifugation for 2 minutes. The resulting solution was diluted 10 times with water before the assay. 10μl of the solution was used in duplicates following dilution with triglyceride assay buffer (up to 50μl). The samples along with Triglyceride standard were incubated with 2μl lipase in 96-well plates for 20 mins at room temperature. Finally, 50μl of master reaction mixture consisting of triglyceride probe, triglyceride enzyme mix and triglyceride assay buffer was added to the samples and incubated at room temperature for 30–60 minutes and the absorbance was measured at 570nm. TAG levels were determined according to the standard curve. Total protein level in the samples were determined by Pierce Bradford Assay kit (Thermo Scientific).

### Glucose quantification

Glucose quantification was performed using glucose assay kit (MAK263, Sigma-Aldrich) from whole bodies of triplicates of 8 males. The males were homogenized in 100μl of PBS. 6μl of the homogenate in duplicates diluted with Assay buffer (up to 50μl) was used for each sample. 50μl of master reaction mix consisting of glucose assay buffer, glucose probe, and glucose enzyme mix was added to the sample and incubated for 30 minutes at 37˚C in dark. Absorbance was measured at 570nm. Glucose levels were determined according to the standard curve. Total protein level in the samples were determined by Pierce Bradford Assay kit (Thermo Scientific) and used for normalization.

### Hydrogen Peroxide quantification

$H_2O_2$ quantification was performed on triplicates of 8 adult males using Amplex Red Hydrogen Peroxide assay kit (A22188, Invitrogen) according to manufacturer's instructions. Briefly, males were homogenized in PBS and centrifuged at high speed to remove any debris. 10ul of the homogenate was diluted with reaction buffer (up to 50ul). A working solution of Amplex red reagent consisting of Amplex reagent stock, HRP and reaction buffer was added to the samples, control, and standards. The reaction mix was incubated for 30 minutes at room temperature protected from light, and absorbance was measured at 560nm. Serial dilution of $H_2O_2$ was used to generate a standard curve.

## Immunostaining

For whole gut immunostaining, appropriately aged guts were dissected in PBS and proceeded according to previously described instructions [44]. Briefly, the dissected guts were immediately fixed in glutamate buffer containing 4% paraformaldehyde for 20 minutes followed by series of methanol washes. The fixed guts were washed in PBS with 0.1% TX-100, followed by blocking in 0.1%TX100 1X PBS containing 5% BSA. Subsequently, the guts were incubated with appropriate antibodies including GFP (1:500), Armadillo (1:100, DHSB), Sox21a (1:5000, a kind gift from Benoit Biteau) and pH3 (1:2000, Invitrogen) overnight at 4˚C. Following day, the guts were incubated in secondary antibody for 2 hours. Images were acquired immediately using confocal microscopy.

## High-fat diet experiments

20-day old males were transferred to vials containing food supplemented with 30% coconut oil for 3 day [45]. For high-fat diet chase experiments, after 3 days on high-fat diet, flies were transferred back to normal food. After 3 days in normal food, gut lipid droplet content was evaluated by BODIPY 493/503 staining.

## Gut motility assay

15 Day old males were raised in food supplemented with 30% coconut oil for 4 days followed by 1 day in high fat food containing Blue-Dye (Blue-1, Spectrum Chemicals, USA). The flies were transferred to normal food and dissected after 8 hours. The guts were counted for retention of blue dye.

## Western blot

Briefly, 10 guts per sample were lysed in 2 X SDS-loading buffer and protein was quantified using Pierce Detergent compatible Bradford Assay kit (Thermo Scientific). The lysates were separated by SDS-PAGE and transferred to Nitrocellulose membrane. The membrane was blocked with 5% non-fat dry milk in TBST and incubated with primary antibodies against Perilipin-2 (1:4000 gift from Dr. Michael Welte) [13,46] and Actin (1:1000, Cell Signaling). HRP conjugated Secondary antibodies were used for detection using SuperSignal West Femto Maximum Sensitivity Substrate (Thermo Fisher, USA). The western blot images were captured by a BIO-RAD workstation.

## Consumption-excretion food assay

As previously described [32], feeder caps containing dyed (Blue-1, Spectrum Chemicals, USA) food media (2%) were placed on vials containing 15 adult males, which were starved for 2 hours. The flies were allowed to consume food for 24 hours. After that the flies were homogenized in water to collect the dye inside (Internal) the flies. The dye excreted by the animals was collected from the walls of the vials using water (External). The absorbance of the internal and external dye was measured by a spectrophotometer at 630 nm. Absorbance values were converted to volume of medium consumed by interpolation of the standard curve for pure dye. Flies fed with food without dye, were used for background absorbance.

## Smurf assay

Appropriately aged males were subjected to Smurf assay as described previously [34]. Briefly, flies (10 males per vial in triplicates) aged on standard media were transferred to food

containing Blue dye no.1 (Spectrum Chemicals, USA) at a concentration of 2.5% for 9 hours. A fly with blue coloration outside of the digestive tract was counted as a Smurf.

### Imaging and signal analysis

For the quantification of the number of pH3+ve cells in the guts, pH3+ve cells were counted manually on a Zeiss Fluorescence microscope.

Confocal images were collected using a Leica SP5 confocal system and processed using the Leica- LAS-X software, analyzed using the Fiji/ImageJ software package and were assembled using Adobe illustrator. Male guts have two prominent regions of lipid accumulation, an anterior region likely corresponding to R2 region and posterior region corresponding to R4/5 regions. All the gut images were captured from the R4/5 region of the midgut. For the quantification of the BODIPY F16 signal, images were analyzed with the Fiji/ImageJ software package. Average pixel intensity in each gut in the preserve lipid droplets was calculated and normalized by subtracting the average pixel intensity in the background region. For the measurement of the area the following sequence was used on Image J. Image> Adjust> Color threshold > select and analyze particles in the image. The percent of total area covered by the lipid droplets was calculated. All the box plots were analyzed using GraphPad Prism 10.0 software. The size of lipid droplets for S2B Fig was measured using Image J/fiji software.

### Lipid incorporation assay

For the incorporation of fluorescently labelled lipid in the lipid droplets, flies were fed with yeast paste containing 1% sucrose and 5mM BODIPY F16 (Invitrogen) for 2 days in dark [47]. The assay was standardized following multiple experiments. The guts were dissected in 1XPBS and fixed in 4% paraformaldehyde for 30 minutes. After fixation the guts were washed twice with PBS and were mounted in 70% glycerol and imaged by confocal microscopy.

### Statistics

All p-values were calculated using the student's t-test with unpaired samples. All error bars represent standard error of mean. Exact values of all $n$'s can be found in Figure legends.

### Dryad DOI

https://doi.org/10.5061/dryad.8sf7m0csb [48]

### Supporting information

**S1 Fig.** (A) Relative expression of *Nazo* transcripts in guts from WT and *nazo* mutant flies (Student's t-test **** p-value = < 0.0001) (B) Standard climbing assay performed on 5-day old male flies of indicated genotypes. Data are mean ± s.e.m. (N = 60 for each genotype) (C) Relative expression of *rp49* (control), *nazo* and *CG3740* transcripts in gut extracts (D) Relative expression of human *c19orf12* in indicated human tissue (image spliced from RNA expression consensus dataset from human protein atlas). (E) Confocal images of whole guts from 20-day old males of indicative genotypes stained with BODIPY 493/503 dye–green and DAPI–blue: Scale bar = 5μm. There are two LD rich regions in male guts both of which exhibit lipid droplets depletion in 20-day old *nazo* mutant males. (F) Confocal images of 20 Day old female guts stained with BODIPY 493/503 dye–green and DAPI–blue: Scale bar = 10μm. (G) Quantification of the percentage area occupied by lipid droplets midguts of flies of indicated genotypes (N = 5–6 females, Student's t-test ** p-value = 0.0074).
(TIF)

**S2 Fig.** (A) Confocal images of male guts of indicated ages and genotypes stained with BOD-IPY 493/503 dye- green and DAPI- blue (Scale bar 10μm). (B) Quantification of the size of lipid droplets in 20-day old male guts of indicated genotypes (N = 20, Student's t-test **** p-value = < 0.0001).
(TIF)

**S3 Fig.** (A) Guts from 20-day-old males with *relish* mutation or ubiquitous knockdown of *dSTING* by *Actin-Gal4* driven *dSTING* RNAi do not show gut lipid droplet depletion on BOD-IPY 493/503 dye staining (Scale bar 10μm). (B) Quantification of the amount of food consumed by adult males in 24-hour period laced with Blue-1 food dye color shows that *nazo* mutants do not have feeding defects. (N = 20, Student t-test p-value ns = not significant) (C) Quantification of gut fluorescence in flies of indicated genotypes fed with fluorescent oleic acid reveals that that *nazo* mutants are not defective in absorption of fluorescent labeled oleic acid (N = 20, Student t-test, p-value ns = not significant). (D) Quantification of whole-body glucose from males of indicated genotypes. 8 males per genotype per biological replicates were used in triplets. (Student t-test p-value ns = not significant) (E) Nazo microRNA leads to efficient depletion of *Nazo* transcript. Relative expression of *nazo* transcripts in guts of flies ubiquitously expressing *Nazo^{mircroRNA}* under *actin-Gal4* driver. The three transgenic lines represent different insertions of the same microRNA transgene. (Student's t-test p-value **** = <0.0001).
(TIF)

**S4 Fig.** (A) Confocal images of guts from 20-day old adults males expressing *Nazo^{microRNA}* under the control of indicated cell type-specific *Gal4* drivers, stained with BODIPY 493/503 dye (Scale bar = 10μm) (B) Quantification of the percentage area covered by lipid droplets in male guts of indicated genotypes.
(TIF)

**S5 Fig.** (A) Relative expression of *nazo* transcript in the guts of WT, *nazo* mutant and *UAS-nazo^{Myc}* rescue flies (*NP-Gal4, UAS-Nazo^{myc}; nazo/nazo*); Student's t-test p-value **** = <0.0001 (B) Standard climbing assay performed on 20-day old male flies of indicated genotypes. Data are mean ± s.e.m. (C) Quantification of the length of the gut in 20-day old males show no significant difference in the overall length in wildtype and *nazo* mutant guts (N = 11) (D) Quantification of the proportion of Smurf male flies in wildtype and nazo mutant males show no significant difference in the gut integrity. (N = 10 males in triplicates) (E) Confocal images of R4/5 regions of the male guts of indicated genotypes and ages stained with Armadillo (Green), Sox21a (Red) and DAPI (Blue) (arrow heads: stem cells and enteroblasts; arrows: enterocytes). (F) Quantification of the number of pH3+ve cells in the guts of wildtype and *nazo* mutant males at indicated ages. Each data point represents a single gut.
(TIF)

**S6 Fig.** (A) Confocal image of fat body stained with BODIPY 493/503 dye from adult males shows that expression of *nazo^{microRNA}* under the ubiquitous driver *Actin-Gal4* leads to diminished lipid droplets in fat body relative to control flies (BODIPY 493/503 dye–green and DAPI–blue; Scale bar = 10μm) (B) Quantification of the percentage area occupied by lipid droplets in the fat bodies of indicated genotypes (N = 5–6 males; Student's t-test *** p-value = < 0.005) (C) Quantification of the TAG levels in whole body of 20-day old males of indicated genotypes reveal that enterocyte but not fat body specific depletion of Nazo leads to diminished TAG levels. 8 males per genotype per biological replicates were used in triplicates. (Student's t-test **** p-value = < 0.0001, ns = not significant. (D) Bright field images of the crops of 20-day old flies of indicated genotypes raised on high-fat diet for 3 days (Scale bar = 5μm).

(E) Quantification of percentage of guts with retained dye in indicated (N = triplicate of 10 guts, Student's t-test *** p-value = 0.0007).
(TIF)

**S7 Fig.** (A) Confocal images of 7-day old male guts stained with BODIPY 493/503 dye and DAPI-blue. Scale bare 10μm. (B) Quantification of lipid droplet area in 7-day old males of indicated genotypes (N = 5–6 males; Student's t-test, ns).
(TIF)

**S8 Fig.** Confocal images of guts from 20-day old flies harboring a *GFP trap* in *plin-2 (LSD-2)* locus stained with GFP (green), Lipid-TOX-Red, and DAPI shows reduced levels of Perilipin-2 upon Nazo depletion (Scale bar = 10μm).
(TIF)

**S9 Fig.** (A) Confocal images of guts from 20-day old flies of indicated genotypes stained with BODIPY 493/503 dye (BODIPY 493/503 dye–green and DAPI–blue; Scale bar = 10μm); Concurrent Knockdown of the lipase *brummer* but not *dob* in *CG740 null*, *nazo microRNA* background using ubiquitous *actin-Gal4* driver can rescue the reduced gut lipid droplet phenotype resulting from Nazo depletion. Similarly, overexpression of Perilipin-2, a regulator of Brummer, but not Sturkopf (another lipid metabolism gene) rescues the nazo depletion phenotype (B) Quantification of the percentage area occupied by lipid droplets in guts of flies of indicated genotypes (N = 5–6 males, Student's t-test **** p-value = <0.0001, ** = 0.002).
(TIF)

**S1 File. List of genes downregulated in guts of *nazo/CG3740* double mutants relative to WT males.**
(XLSX)

**S2 File. List of genes upregulated in guts of *nazo/CG3740* double mutants relative to WT males.**
(XLSX)

**S3 File. GO molecular function analysis of upregulated and downregulated genes from the gut of WT vs *nazo/CG3740* double mutants.**
(XLSX)

**S4 File. Western blot for HA-Nazo Perilipin-2 interaction.**
(XLSX)

**S5 File. Quantification of numerical values for imaging studies described in the study.**
(XLSX)

## Acknowledgments

RNAseq was performed at the University of Rochester Genomics Research Center.

## Author Contributions

**Conceptualization:** Perinthottathil Sreejith, Leo J. Pallanck, Rajnish Bharadwaj.

**Data curation:** Perinthottathil Sreejith, Sara Lolo, Kristen R. Patten, Neya More, Rajnish Bharadwaj.

**Formal analysis:** Perinthottathil Sreejith, Sara Lolo, Kristen R. Patten, Maduka Gunasinghe, Neya More, Leo J. Pallanck, Rajnish Bharadwaj.

**Funding acquisition:** Leo J. Pallanck, Rajnish Bharadwaj.

**Investigation:** Perinthottathil Sreejith, Sara Lolo, Kristen R. Patten, Maduka Gunasinghe, Neya More, Leo J. Pallanck, Rajnish Bharadwaj.

**Methodology:** Perinthottathil Sreejith, Sara Lolo, Kristen R. Patten, Maduka Gunasinghe, Neya More, Rajnish Bharadwaj.

**Project administration:** Rajnish Bharadwaj.

**Resources:** Rajnish Bharadwaj.

**Supervision:** Leo J. Pallanck, Rajnish Bharadwaj.

**Validation:** Perinthottathil Sreejith, Sara Lolo, Kristen R. Patten.

**Visualization:** Perinthottathil Sreejith, Sara Lolo, Kristen R. Patten, Maduka Gunasinghe, Leo J. Pallanck, Rajnish Bharadwaj.

**Writing – original draft:** Perinthottathil Sreejith, Rajnish Bharadwaj.

**Writing – review & editing:** Perinthottathil Sreejith, Leo J. Pallanck, Rajnish Bharadwaj.

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
