## [Decision Letter · Decision Letter 0]

1 Sep 2023

Dear Dr Bharadwaj,

Thank you very much for submitting your Research Article entitled 'Nazo, the Drosophila homolog of the NBIA-mutated protein – c19orf12, is required for triglyceride homeostasis' to PLOS Genetics.

The manuscript was fully evaluated at the editorial level and by independent peer reviewers. The reviewers appreciated the attention to an important problem, but raised some substantial concerns about the current manuscript. Based on the reviews, we will not be able to accept this version of the manuscript, but we would be willing to review a revised version. We cannot, of course, promise publication at that time.

If you decide to revise the manuscript for further consideration at PLOS Genetics, please aim to resubmit within the next 60 days, unless it will take extra time to address the concerns of the reviewers, in which case we would appreciate an expected resubmission date by email to plosgenetics@plos.org.

We are sorry that we cannot be more positive about your manuscript at this stage. Please do not hesitate to contact us if you have any concerns or questions.

Yours sincerely,

Pablo Wappner

Academic Editor

PLOS Genetics

Gregory P. Copenhaver

Editor-in-Chief

PLOS Genetics

Reviewer's Responses to Questions

**Comments to the Authors:**

Reviewer #1: The manuscript uncovers the role of Drosophila Nazo, the homolog of c19orf12 - a protein implicated in the neurodegenerative disorder NBIA. Using a combination of genetic, biochemical, and RNAseq studies, they find that Nazo is critical for lipid droplet accumulation in the gut. Nazo regulates lipid droplets by controlling the levels of Perilipin 2, which are critical for inhibiting the lipid droplet associated lipase Brummer. Specifically, loss of Nazo decreases Perilipin 2 levels, increasing Brummer activity and lipid droplet lipolysis. This makes mutants for nazo sensitive to starvation and oxidative stress. This study provides key insights into the cellular functions of Nazo, and will be of interest to researchers in the areas of lipid droplet biology, gut homeostasis, and neurodegeneration. However, there are a number of areas where the study can be improved.

Major

1. The statement that LDs must remain in contact with the ER is incorrect and lack of contact does not always result in LD defects. Lines 67-71.

2. The description of the effect of Seipin is also not accurate. Loss of Seipin also contributes to reduced LD biogenesis which could drive the phenotype (line 72-73)

3. There are a number of instances where the conclusions/outcomes are overstated.

a. Line 1130-114. The study does not study the functions of C19orf12, so it cannot provide insight into the molecular basis of NBIA. The simple addition of “potentially” would address this concern.

b. Line 127-128: saying the Cg3740 has an effect on longevity is inaccurate given no significant differences.

c. Lines 241-242 – the statement that Nazo overlaps with ER is not complete accurate. Some foci do overlap, but some appear not to. Also there is not quantification of the localization – in particular Nazo in LD-ER contact sites vs not.

d. Lines 294-295 – the statement that Nazo is a membrane contact site protein is an overstatement. It colocalizes with/near ER and LDs, but that doesn’t show it is a membrane contact site protein.

4. Figure panels are not presented in the order discussed in the text – this is mostly in relation to panel order on the supplemental figures.

5. In the figures the order in which nazo vs Cg3740 data is presented is inconsistent. It would be helpful to the reader to be able to see the data for the two mutants in the same order every time.

6. It is unclear in the results/legends when male or female flies are used. Further in the methods it is unclear why some experiments only use males.

7. What region of the midgut is used for the images/analyses?

8. Given the lack of/mild phenotypes at day 7 and the striking defects at day 20 – it would interesting to determine when the defects arise. Also the age dependency of the phenotype needs addressed in the discussion – why do the authors think this is age dependent? How does that relate to the disease.

9. The LD phenotype seems more complex than just area of LDs in the cells, the LD size seems dramatically different – something seem in the Day 7 images.

10. It is unclear what was done in Fig S3C – was the fluorescent lipid incorporated into the same cellular structures in the control vs mutant? (ie LDs?)

11. Fig. S4a – lines 189-192 – the statement of no ‘significant’ differences requires quantification. Further, the LD numbers look reduced (not as severely as with NP).

12. Missing controls for the rescue experiment. It is important to see what GAL4; nazo phenotypes are vs GAL4; nazo; UAS – or at least the GAL4 by itself.

13. The CG3470 sensitivity to starvation by little change in LDs is interesting. It is worth discussing why this is the case. Stats between CG3470 and nazo would also add to the findings.

14. Lines 230-231 – the data for the dye gut mobility assay should be shown.

15. S2 cell localization images are hard to interpret. LDs are massively overexposed. Merge images of just ER and Nazo, and LD and Nazo are needed.

16. ER staining in the gut (Fig. 6B) is not convincing.

17. Missing stats in Fig 8A-B.

18. Western blots – full western blots should be provided in the supplemental figures. More than one sample needs shown, and/or quantification is needed.

19. The description of the data in the results for SFig 7 doesn’t indicate that only double mutants were used. It is also not clear why that is the genotype being studied.

20. The NBIA phenotype has iron accumulation, the RNAseq data in the manuscript has genes misregulated that are involved in metal metabolism. Iron is critical for gut homeostasis. This needs addressed in the discussion.

21. A missing and important piece of information is whether the gut morphology, cell type distribution is impacted by loss of Nazo.

22. Lines – 375-377 – PLIN2 and nazo don’t interact. Where is this data in the paper?

23. Does the enterocyte rescue of Nazo restore lifespan and climbing ability?

24. The model is not clear – some indication of lipolysis is needed, and perhaps boxing/labeling the different contexts.

Minor

1. Abbreviations should not be used in the abstract (NBIA) without being defined.

2. Typo Line 34 – function should be functional

3. Inconsistent use of Drosophila melanogaster vs Drosophila, and sometimes italicized and sometimes not.

4. The LDs in the relish mutant and Sting RNAi don’t look normal – this is important to mention.

5. Fig 3B – the stats marks are misplaced.

6. Missing figure callout – Line 285-86

7. Paragraph starting at line 393 doesn’t seem appropriate for the discussion. Instead adding how gut phenotypes may contribute to NBIA would be useful – are patients starvation/oxidative stress sensitive – is this driving neurodegeneration?

8. Missing wavelength of analysis – line 611

9. Seems like H2O2 measurements in Cg3470 are warrented – all other experiments assay both it and Nazo.

Other

- The experiment in Fig 7A-B is very informative and convincing!

Reviewer #2: The manuscript from Sreejith et al attempts to delineate the function of c19orf12 utilizing a drosophila model system and its two c19orf12 homologs. The authors show that one homolog – Nazo – impacts gut lipid metabolism and suggest this occurs via alterations in lipolysis. The authors further suggest important roles for the lipid droplet protein PLIN2 and lipase ATGL as mediating the downstream effects. These results suggest new roles for Nazo. Yet, there are significant concerns with various aspects of this manuscript as listed below.

• Throughout the manuscript, gal4 is typically used as the sole control. However, uas may also be influencing the phenotype. Some additional controls including a uas driver or inducible system are needed to show the effects are true and not merely an artifact of high gal4 overexpression.

• The authors only show a few genes that are changed with their RNAseq analysis. Despite stating pathway analysis, heatmap generation, etc. in the methods, no pathway analysis is shown.

• A major concern is the lack of quantification of the manuscript. The images and western blot all appear to be an n=1 with no quantification.

• The central premise of this manuscript is that lipolysis is a key mediator of Nazo. However, lipolysis is never measured. Changes are inferred. The data attempting to dissect incorporation vs breakdown explaining the lipid droplet phenotype is not convincing as there are no kinetic measurements across time. Cell-based studies are needed to tease this apart.

• The conclusions showing that Nazo is localized to ER-LD contact sites are not convincing. Higher-resolution microscopy with quantification including cell-based studies is needed.

• The findings that the crops are robustly enlarged could explain significant differences in gut lipid metabolism. It is unclear why this was mentioned but not further studies.

• Coconut oil is highly enriched in medium chain fatty acids, which are metabolized very differently than the long chain fatty acids found in most fat sources. It is unclear how to interpret this data given the difference in medium chain absorption and metabolism.

•

Minor

• To improve readability, please put the figures in the order in which they are introduced in the text – i.e. S1a, S1b, etc.

• It is disappointing that only males were studied given that females and males have well-known differences in lipid metabolism.

• S2A is incorrectly referenced in the text.

• Nazo expression should be measured in the rescue studies.

• Some of the figures don’t go well together – e.g. the oxidative stress and crop data don’t appear to be connected.

Reviewer #3: In this paper, Sreejith et al provide a characterization and study of nazo, the fly homolog of human c19orf12, a neurodegenerative disease-associate gene. They show how nazo function, specifically in the intestinal enterocyte cells, is needed to maintain both gut and whole-body levels of TAGs. Using a newly generated mutant, they show that loss of nazo confers stress sensitivity to flies. Finally, they show that nazo localizes to ER-lipid droplet sites and they find that the lipid loss seen in nazo mutants can be reversed by loss of the lipase bmm, suggesting nazo may play a role in regulating lipolysis.

Overall, this is a solid paper. By identifying a new regulator of lipid metabolism, this study will be of interest to researchers interested in metabolism in flies and other systems. One drawback of the work is that it doesn’t actually pinpoint what the molecular role of nazo is, hence overall the study lacks mechanistic depth.

I have the following specific comments that the authors should address before publication

1, the study is focused on experiments in males only. The authors should explain and rationalize this choice. They do refer to one result (not presented) that showed that the lipid depletion phenotype is not as strong in females as in males. This result should be shown and quantified. There is an increasing emphasis and interest in sex differences in metabolism, and male-female differences in fly lipid metabolism have been shown (e.g., Wat et al, eLife, 2021). Hence, I think it would be good to report the result with female nazo mutants.

2, The authors should also explain and rationalize their choice of using 20-day old male flies. At this time point, male flies are approaching the point of onset of age-related phenotypes especially with regard to lipid metabolism and stress tolerance (e.g, Karpac, Biteau, Jasper, Cell Reports, 2013). Indeed, the authors show one result with 7d flies suggesting the nazo mutant lipid depletion phenotype is not as strong. This result should be quantified and reported. If the nazo phenotype are age dependent, this will be important to confirm and report, to help interpret and consider the biological roles for nazo.

3, The authors should provide a little more detail about how the quantifications of lipid staining were performed. In Fig 8A, some the lipid droplets seem larger in size that the actual cells (based on nuclei distributions).

**Have all data underlying the figures and results presented in the manuscript been provided?**

Reviewer #1: **No: **Full western blots, excel sheets of image quantifications

Reviewer #2: Yes

Reviewer #3: Yes

PLOS authors have the option to publish the peer review history of their article (what does this mean?). If published, this will include your full peer review and any attached files.

Reviewer #1: No

Reviewer #2: No

Reviewer #3: No

---

## [Decision Letter · Decision Letter 1]

28 Dec 2023

Dear Dr Bharadwaj,

Thank you very much for submitting your Research Article entitled 'Nazo, the Drosophila homolog of the NBIA-mutated protein – c19orf12, is required for triglyceride homeostasis' to PLOS Genetics.

The manuscript was fully evaluated at the editorial level and by two of the reviewers that analyzed the original submission (reviewers #2 and #3). One of the reviewers (#3) was satisfied with the revissions, while the reviewer #2 raised relatively minor issues that needs to be addressed. The responses to the concerns of reviewer #1 were analyzed at the editorial level, and considered satisfactory. We noticed however that supplementary figure 4 is mislabled (two Sup Figures 3 appear instead). Please correct this when you resubmit the manuscript.

We therefore ask you to modify the manuscript according to reviewer #2 recommendations. 

1) Provide a detailed response to the reviewer comments and a description of the changes you have made in the manuscript.

Yours sincerely,

Pablo Wappner

Academic Editor

PLOS Genetics

Gregory P. Copenhaver

Editor-in-Chief

PLOS Genetics

Reviewer's Responses to Questions

**Comments to the Authors:**

Reviewer #2: There seems to be an image in Figure 3 missing (4 images and 5 quantifications for C-E).

Please check the scale bars in figure 5. It states they are 100 microns but that would make these the largest S2 cells to ever exist.

I am still struggling with the imaging data for where Nazo is localized. I understand the difficulties of imaging the gut, but I simply don't think the resolution and quality of images in both S2 or guts are convincing enough to even say it is an LD-associated protein. I think there is enough solid data in this manuscript that the localization and mechanisms of how nazo regulates lipolysis should be left and further explored in more depth in future studies.

Reviewer #3: The revised manuscript address all my previous concerns and I am happy to recommend acceptance of the paper

**Have all data underlying the figures and results presented in the manuscript been provided?**

Reviewer #2: Yes

Reviewer #3: Yes

PLOS authors have the option to publish the peer review history of their article (what does this mean?). If published, this will include your full peer review and any attached files.

Reviewer #2: No

Reviewer #3: No

---

## [Editor Report · Decision Letter 2]

12 Jan 2024

Dear Dr Bharadwaj,

We are pleased to inform you that your manuscript entitled "Nazo, the Drosophila homolog of the NBIA-mutated protein – c19orf12, is required for triglyceride homeostasis" has been editorially accepted for publication in PLOS Genetics. Congratulations!

Yours sincerely,

Pablo Wappner

Academic Editor

PLOS Genetics

Gregory P. Copenhaver

Editor-in-Chief

PLOS Genetics

Comments from the reviewers (if applicable):

**Data Deposition**

http://datadryad.org/submit?journalID=pgenetics&manu=PGENETICS-D-23-00784R2

**Press Queries**

---

## [Editor Report · Acceptance letter]

6 Feb 2024

PGENETICS-D-23-00784R2 

Nazo, the Drosophila homolog of the NBIA-mutated protein – c19orf12, is required for triglyceride homeostasis 

Dear Dr Bharadwaj, 

We are pleased to inform you that your manuscript entitled "Nazo, the Drosophila homolog of the NBIA-mutated protein – c19orf12, is required for triglyceride homeostasis" has been formally accepted for publication in PLOS Genetics! Your manuscript is now with our production department and you will be notified of the publication date in due course.

With kind regards,

Zsofi Zombor

PLOS Genetics

On behalf of:
